# Self-derived Knowledge Graph Contrastive Learning for Recommendation

## ABSTRACT

Knowledge Graphs (KGs) serve as valuable auxiliary information to improve the accuracy of recommendation systems. Previous methods have leveraged the knowledge graph to enhance item representation and thus achieve excellent performance. However, these approaches heavily rely on high-quality knowledge graphs and learn enhanced representations with the assistance of carefully designed triplets. Furthermore, the emergence of knowledge graphs has led to models that ignore the inherent relationships between items and entities. To address these challenges, we propose a Self-Derived Knowledge Graph Contrastive Learning framework (CL-SDKG) to enhance recommendation systems. Specifically, we employ the variational graph reconstruction technique to estimate the Gaussian distribution of user-item nodes corresponding to the graph neural network aggregation layer. This process generates multiple KGs, referred to as self-derived KGs. The self-derived KG acquires more robust perceptual representations through the consistency of the estimated structure. Besides, the self-derived KG allows models to focus on user-item interactions and reduce the negative impact of miscellaneous dependencies introduced by conventional KGs. Finally, we apply contrastive learning to the self-derived KG to further improve the robustness of CL-SDKG through the traditional KG contrast-enhanced process. We conducted comprehensive experiments on three public datasets, and the results demonstrate that our CL-SDKG outperforms state-of-the-art baselines.

## CCS CONCEPTS

• **Information systems** → **Recommender systems**.

## KEYWORDS

Recommendation system, contrastive learning, Knowledge graph

## 1 INTRODUCTION

Recommendation systems[16, 21, 50] play a crucial role in helping online users discover relevant information by suggesting items that align with their interests. Considering the interaction between users and items as first-order neighbors of users in the user-item graph neural networks (GNNs)[18, 24, 27, 58] have become widely employed in the field of recommendation. Numerous studies have

*ACM MM, 2024, Melbourne, Australia*
© 2024 Copyright held by the owner/author(s). Publication rights licensed to ACM.
ACM ISBN 978-x-xxxx-xxxx-x/YY/MM
https://doi.org/10.1145/nnnnnnn.nnnnnnn

**Unpublished working draft. Not for distribution.**

demonstrated that GNNs can capture high-order implicit connectivity in user-item interaction graphs, thereby improving the performance of collaborative filtering [31, 42, 47]. However, when dealing with a vast user-item interaction diagram [13, 35], unrelated items may obscure the user's genuine interests. In an attempt to mitigate this selection error, some researchers often utilize message drop techniques [2, 5, 7], randomly discarding outbound messages based on the user-item diagram. Nevertheless, this approach can lead to a sparser user-item interaction graph in extremely high sparsity recommendation scenarios [15, 17, 23, 53], resulting in diminished robustness of GNNs.

Subsequently, contrastive learning (CL) [19, 54, 55] offers a self-supervised manner to learn user and item embeddings, demonstrating its effectiveness in enhancing the robustness of GNNs. Specifically, for each node in the user-item graph, its subgraph undergoes random perturbation, involving the random discarding of a subset of its subgraph edges. Notably, techniques like SGL [38] and SLRec [46] leverage contrastive self-supervised tasks to improve node representations in GNNs through self-discrimination [39, 40, 49]. Then VGCL [45] samples the input graph to reconstruct it and performs contrastive learning at both node and cluster levels. Finally, CGCL [9] takes similar semantic structure representations to design contrastive learning at three levels: neighbor structure, candidate structure, and candidate neighbor, resulting in high-quality node representations. However, in the process of generating contrast subgraphs, the above methods disrupt the original graph structure, potentially erasing valuable information and consequently diminishing the representational power of nodes.

Recently, researchers have turned to Knowledge Graph (KG) assisted GNNs [1, 6, 20, 25] to elevate the performance of recommendation systems, which can overcome the potential disruption caused by contrastive learning and enhance node representations. Utilizing a relationship graph converter, RGTN [14] propagates information between nodes and employs various encoders to capture both the structural and semantic features of nodes. Moreover, it uses the attention fusion module to obtain the importance values of knowledge graph nodes. Notably, KGCL [44] addresses noise in knowledge graphs by incorporating additional supervised signals during the KG enhancement process, guiding cross-view contrastive learning paradigms that play an important role in unbiased user-item interactions during gradient descent. Additionally, When KGRec [43] conducts contrastive learning, it maintains consistency between the two signals of knowledge and user-item interaction, which employs the masking of important knowledge through high rationalization scores. Despite these strengths, current KG-based methods heavily depend on the quality of the used KG. Moreover, an excessive reliance on KG may introduce a learning bias, where models become dominated by KG features, potentially overlooking valuable user-item connections in the original graph.

To address the challenges mentioned above and achieve excellent performance and robustness, we present the Self-Derived subgraphs

Contrastive Learning framework (CL-SDKG). In the initial step, we use the graph from each GNN layer as input for the variational graph reconstructor, allowing us to estimate the Gaussian distribution of user-item nodes corresponding to the GNN aggregation layer. This innovative approach enables the generation of multiple knowledge graphs, referred to as self-derived knowledge graphs, which can reduce undue reliance on the knowledge graph and mitigate noise associated with it through the consistency of the estimated structure. Moreover, the self-derived knowledge graphs not only facilitate a better understanding of relationships between user-item nodes but also serve as a mechanism for the model to comprehend and utilize valuable connections within the original graph. This unique feature empowers the CL-SDKG method to comprehensively leverage information from the original graph, avoiding biases toward knowledge graphs. Additionally, we incorporate signals during the original knowledge graph contrastive enhancement process to conduct contrastive learning on the self-generated subgraphs. This step further fortifies the model's robustness, enabling it to adapt more effectively to various user-item interaction scenarios and suppress noise during gradient descent.

In summary, the main contributions of our paper are:

- We propose a novel recommendation algorithm CL-SDKG, which has excellent robustness and better recommendation performance, i.e., it can effectively reduce the noise information brought by knowledge graphs and reduce excessive dependence on knowledge graphs.
- We use a variational estimator to reconstruct the input graph as a learnable version of the traditional knowledge graph, which can effectively correct and supervise the model's behavior during the learning process and enable the model to learn effective knowledge connections of the data, without overly relying on the traditional knowledge graph.
- We validated the effectiveness of CL-SDKG on three real-world datasets. The experimental results strongly indicate that CL-SDKG outperforms other advanced methods in terms of performance.

## 2 RELATED WORK

### 2.1 GNN in Recommendation

Recommendation systems are increasingly gaining popularity among users seeking personalized items. Traditionally, Relying on historical user-item interactions to uncover user preferences and interests, collaborative filtering models [30, 56] have dominated the realm of recommendation systems. However, these methods encounter challenges stemming from intricate user behavior or data input complexities. In response to these challenges, GNN-based recommendation methods have garnered the attention of researchers. Graph recommendation methods have proven to deliver impressive recommendation performance, primarily owing to their adeptness in effectively leveraging graph structures to capture meaningful connections between nodes during the iterative process. For instance, AutoGCL [52] utilizes multiple graph view generators to learn the probability distribution of input graphs. And it also introduces sufficient enhanced variance in the contrastive learning process. To generate CL views, XSimGCL [48] utilizes invalid graph enhancements, employing a simple and effective noise-based

embedding approach. LightGCN [10] enhances recommendation performance by eliminating nonlinearity and obtaining the initialization of embeddings by computing the network embedding on the compressed graph for each node. This approach also alleviates the over-smoothing phenomenon associated with sparse interaction data in GNNS. However, the high sparsity of the user-item interaction graph presents a challenge, as GNNs may lack sufficient generalization ability, leading to an ineffective perception of the dependency relationship between users and items.

### 2.2 Knowledge Graph

Enhanced knowledge graph recommendation involves two ways: embedding enhancement and path enhancement. In the embedded recommendation methods, the emphasis is on leveraging the auxiliary information inherent in the knowledge graph. For instance, Kopra [29] derives the corresponding user representation by pinpointing relevant entities in the knowledge graph based on the user's click history. CKE [51], On the flip side, captures both structural and semantic representations of items by extracting information from the heterogeneity of nodes and the knowledge graph (KG), covering aspects such as structure, text, and vision. In a similar vein, KGTORE [26] employs the knowledge graph to learn potential representations of semantic features. This enables the interpretation of user decisions as a personal sublimation of item feature representations.

Enhancing path recommendation methods primarily focus on the representations of meta-paths. In their work, [12] introduces a three-way neural interaction model utilizing priority sampling techniques to select better path instances. The model is based on meta-paths, enhancing the representation ability of context, user, and item, with the three elements reinforcing each other. RippleNet [32] automatically extends users' potential interests along links in the KG, encouraging the propagation of user interests among knowledge entities. In a similar vein, KGCN mines pertinent attributes on the knowledge graph to efficiently capture correlations between items. While path-based recommendation methods demonstrate superior performance compared to embedding-based methods, as they can capture higher-order knowledge perception dependencies, they heavily rely on domain knowledge-based meta-path design. Additionally, the exploration of different meta-paths involves significant time costs.

### 2.3 Graph Contrastive Learning

Graph Contrastive Learning (GCL) has emerged as a prominent trend in recent years within the research community. Many GCL approaches [33, 38] incorporate additional supervisory signals from raw data for recommendation. [44] introduces a universal Knowledge Graph Contrastive Learning framework (KGCL) to reduce noise information within KG-enhanced recommendation systems. This approach utilizes the knowledge graph expansion pattern to mitigate noise during information aggregation, leading to a more resilient representation of item perception. KGRec [43] introduces a novel self-supervised rationalization method, producing rationalization scores for knowledge triplets. Specifically, it integrates generative and contrastive self-supervised tasks to enhance model reconstruction by assigning high rationalization scores to crucial

knowledge while masking it. However, GNNs combined with KG recommendation methods [34, 36]heavily rely on the quality of knowledge graphs. Moreover, excessive reliance on knowledge graphs may cause the model to learn features biased towards the KG, potentially overlooking useful connections between nodes in the original graph. In comparison, our method CL-SDKG achieves superior performance and robustness by utilizing self-generated subgraphs from the original graph for contrastive learning.

## 3 PRELIMINARIES

In this section, we mainly introduce the key symbols and notations used in our paper and formalize the definitions of our research problem. We adopt $\partial = (\mathbf{U}, \mathbf{V})$, $u \in \mathbf{U}$ and $v \in \mathbf{V}$ represent individual users and items, respectively. We construct a user-item interaction diagram $\mathbf{G_u} = (u, I, v)$ to represent whether the user and the item have the interaction. Here, the presence of an interaction between a user and an item is expressed as $I = 1$. Conversely, the absence of interaction is denoted as $I = 0$. We use knowledge graphs containing triples to represent the actual knowledge of related items, represented as $\mathbf{G_{kg}} = (e_1, r_{e_1,e_2}, e_2), e_1, e_2 \in \mathbb{E}^m$, where $\mathbb{E}$ is the $m$-dimensional vector set of knowledge entities, $r_{e_1,e_2}$ represents the relationship between two entities such as (engineer, build, house). It's crucial to emphasize that the entity set is not exactly equal to the item set. Indeed, it is a subset of the entity set, enabling the reconstruction of meaningful connections between items and entities within the knowledge graph.

We employ a variational estimator $F(\mathbf{G_{kg}^i}|u, v, \mathbf{G_u^i}, \mathbf{G_{kg}})$, to estimate the distribution of nodes corresponding to the iterated user-item graph. This process yields multiple KGs, represented as $\mathbf{G_{kg}^i}^*$. Leveraging these KGs enhances our model's ability to learn the representation of the input graph, resulting in heightened robustness.

Finally, our task can be formally defined as follows: considering a user-item interaction graph denoted by $\mathbf{G_u}$, and a knowledge graph represented by $\mathbf{G_{kg}}$, Our objective is to develop a recommendation model $H(u, v|\mathbf{G_u}, \mathbf{G_{kg}}, \mathbf{G_{kg}^i}^*), \omega)$, where $\omega$ represents learnable parameters. The model outputs a value within the range of $[0, 1]$, indicating the likelihood of user $u$ interacting with item $v$.

## 4 METHODOLOGY

In this section, we will provide a detailed overview of CL-SDKG. The overall framework of our CL-SDKG is illustrated in Figure 1. The CL-SDKG comprises three main parts: (1) Knowledge aggregation enhanced recommendation. (2) Variational graph reconstruction and enhanced representation of self-derived KGs. (3) Contrastive learning of self-derived KGs in user-item view. For initialization, we use free embeddings $u \in \mathbf{U}$ and $v \in \mathbf{V}$ to represent user and item nodes. The vectors $u_i$ and $v_j$ correspond to user $i$ and item $j$ respectively. Subsequently, we will explore each component for a more comprehensive understanding.

### 4.1 Knowledge Aggregation Enhanced Recommendation

In dealing with a complex knowledge graph, we employ a fundamental principle involving the weighted learning of the probability of knowledge triplet existence and the basic tenet of collaborative

interaction. The function applies a weighted graph attention mechanism to each knowledge triplet, incorporating learnable content:

$$G_{kg}\left(e_1, r_{e_1,e_2}, e_2\right) = \frac{e_1 \mathbf{W^Q} * \left(e_2 \mathbf{W_k} \odot r_{e_1,e_2}\right)}{\sqrt{d}} \quad (1)$$

where $e_1$, $r_{e_1,e_2}$, and $e_2$ are entity representations of the head, relationship and tail. $\mathbf{W^Q}$ and $\mathbf{W_k}$ are the trainable weights of attention, with a size of $d \times d$. To capture the relevant entities, we rotate $e_1$ to the potential space of $e_2$ by using the relationship $r_{e_1,e_2}$. We use a knowledge triplet with a basic principle score of $\mathbf{G_{kg}}(e_1, r_{e_1,e_2}, e_2)$ to assist the recommendation system in enhancing recommendations. To ensure comparability of the scores, we normalize the scores with the following formulas:

$$\omega\left(e_1, r_{e_1,e_2}, e_2\right) = \frac{\exp\left(\mathbf{G_{kg}}\left(e_1, r_{e_1,e_2}, e_2\right)\right)}{\sum_{e_1, r'_{e_1,e_2}, e'_2 \in \tau_{e_1}} \exp\left(\mathbf{G_{kg}}\left(e_1, r'_{e_1,e_2}, e'_2\right)\right)} \quad (2)$$

where $\tau_{e_1}$ is the is the neighbors of $e_1$, $\omega\left(e_1, r_{e_1,e_2}, e_2\right)$ is the normalized scores, $e'_2$ is the entity connected to $e_1$, and $r'_{e_1,e_2}$ is the relations between $e_1$ and $e'_2$.

### 4.2 Variational Graph Reconstruction and Enhanced Representation of Self-derived Knowledge Graph

In the process of reconstructing variational graphs, we use VAE to estimate the Gaussian distribution of nodes and edges in the input graph to achieve graph reconstruction, thereby generating new subgraphs.

**VAE Brief**. Given a training set, the variational autoencoder [22] operates under the assumption that each sample $x_i$ is part of it. Samples are constructed through a random process, where $x \sim p_\delta(x|k)$. Additionally, we can derive the maximum likelihood function:

$$\log p(x) = \log \int p_\delta(x|k)p(k)\mathrm{d}k \quad (3)$$

where $p(k)$ is the prior distribution of potential variables $k$. However, since it is impossible for us to know all potential variables, VAE adopts variational inference [8] and chooses to use inference models $q_\sigma(k|x)$ to model posterior distributions $p_\delta(x|k)$. Furthermore, variational autoencoders are optimized by minimizing evidence-based lower bounds to obtain the best results:

$$Loss_{ELBO} = -E_{k \sim q_\sigma(k|x)}\left[\log(p_\delta(x|k))\right] + KL[q_\sigma(k|x)||p(k)] \quad (4)$$

where $q_\sigma(k|x)$ and $p_\delta(x|k)$ represent the encoder and decoder of the parameterized neural network; KL is the Kullback-Leibler divergence [41] between $q_\sigma(k|x)$ and $p(k)$, introducing constraints to align it with the vicinity of a prior Gaussian distribution.

Then, we conveniently use Gaussian distribution of input graph nodes and edges for graph reconstruction.

**Graph Reconstruction**. We aim to refactor the input user-item interaction diagram and the initialized node representation. The probability distribution of the input graph structure $k : y' \sim p_\delta(y|k)$ is learnable. Each node in the input graph is encoded as a Gaussian distribution $q_\sigma(k|y, E_\lambda) = N(z_i|\mu_\phi(i), diag(\sigma_\phi^2(i)))$, where $\mu_\phi(i)$ and $\sigma_\phi^2(i)$ represent the mean and variance of the corresponding nodes, respectively. To leverage the deep representation

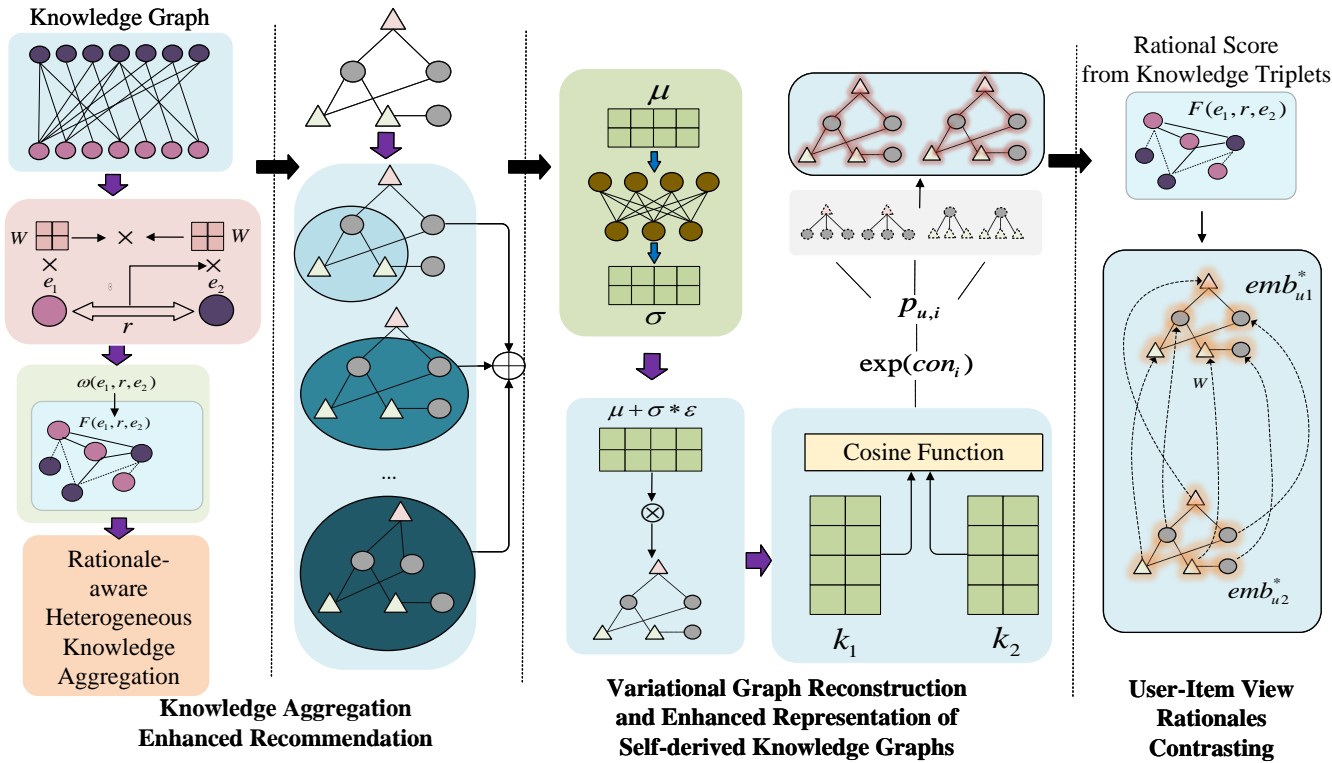

**Figure 1: The overall framework of CL-SDKG.**

of user-item graphs more effectively, we estimate the mean and variance of node distributions by utilizing GNN. After stacking $N$ multiple convolutional layers, we fuse the calculated mean and utilize a Multi-Layer Perceptron (MLP) to calculate its variance. Once the approximate posterior mean and variance are obtained, we generate potential subgraphs $k_i$ by sampling in $N(\mu_i, \sigma_i^2)$. Since this sampling process is non-differentiable, we adopt the reparameterization techniques to replace the sampling process:

$$\mu = \frac{1}{L}\sum_{n=1}^{N}\mu_n \tag{5}$$

$$\sigma = MLP(\mu) \tag{6}$$

$$k_i = \mu_i + \sigma_i * \xi \tag{7}$$

where $\xi \sim N(0, 1)$ is the noise that follows a normal distribution. Once the probability distribution of variables is estimated, our primary objective is to reconstruct the original graph. Therefore, we employ the inner product to calculate the probability score of nodes $i$ and $j$ for connections:

$$p(\text{intr}_{ij} = 1|k_i, k_j) = \text{sigmoid}(k_i^\top, k_j) \tag{8}$$

where $\text{intr}_{ij}$ indicates whether there is interaction between nodes. After deriving the subgraph $\zeta_1, \zeta_2$, we defined the consistency $\text{con}_i$ of the item's knowledge graph structure based on the consistency between graph encoded representations:

$$\text{con}_i = s(f_\zeta(x_i, \zeta_1), f_\zeta(x_i, \zeta_2)) \tag{9}$$

where $f_\zeta(\cdot)$ represents the knowledge aggregation function for relationship perception, and $s(\cdot)$ is the cosine function for estimating the similarity between them. The consistency of the derived knowledge structure can serve as a guide for each item, avoiding certain deviations in the derived knowledge graph and alleviating user-item interaction noise caused by random values of normal distribution.

After obtaining the self-derived knowledge graph comparison subgraph corresponding to each layer, we process it as a knowledge graph. We can predict user interests more effectively by using higher consistency in the knowledge graph structure to reduce noise. To reduce noise in the subgraph, we entails identifying and eliminating weakly correlated edges. Additionally, we incorporate $\text{con}_i$ into the data augmentation process for the user-item interaction graph to mitigate the limitation of contrastive learning due to pure drop operations. The specific formula for knowledge-guided enhancement is as follows:

$$w_{u,i} = \exp(\text{con}_i)\,; p'_{u,i} = \max\left(\frac{w_{u,i} - w_{\min}}{w_{\max} - w_{\min}}, p_\tau\right) \tag{10}$$

$$p_{u,i} = p_x \bullet \mu_p \bullet p_{u,i} \tag{11}$$

where $p_{u,i}$ denotes the probability of the edge connected to the discarded user $u$ and item $i$. $w_{u,i}$ represents the attractiveness of item $i$ to user $u$. $\text{con}_i$ is the structural consistency score corresponding to item $i$. Then we normalize the maximum and minimum truncation probabilities of $w_{u,i}$, where $p_x$ is used to influence the strength of the mean. Subsequently, we generate two mask vectors $mask_u^1$,

$mask_u^2$ based on the Bernoulli distribution:

$$\phi(\mathbf{G_u}) = (\partial, M_u^1 * V), \phi(\mathbf{G_u}) = (\partial, M_u^2 * V) \tag{12}$$

where $f_\phi(\cdot)$ represents the operational symbols that enhance the input graph and discard user-item interactions from the edge set $V$ based on inferred probabilities $p_{u,i}$.

## 4.3 User-item View Rationales Contrasting

After obtaining the enhanced self-derived knowledge graph, we aim to preserve interactive connections that distinctly reflect user interests with high rationality and effectiveness. To achieve this, we opt to assign weights to each interaction edge based on the average rationality score of the knowledge triplet from the original knowledge graph. This approach effectively mitigates the impact of interaction noise.

$$v = mean(F(e_1, r, e_2)) \tag{13}$$

$$\begin{aligned} F(e_1, r, e_2) &= |\tau_{e_1}| \cdot \omega(e_1, r, e_2) \\ &= \frac{|\tau_{e_1}| \cdot \exp(f(e_1, r, e_2))}{\sum_{(e_1, r', e_2') \in \tau_{e_1}} \exp\left(f\left(e_1, r', e_2'\right)\right)} \end{aligned} \tag{14}$$

where the lower $v$ indicates a weaker correlation between the knowledge entities adjacent to the item and the corresponding recommendation task in the knowledge graph. This correlation may introduce deviations to the model. Consequently, we filter out these weak connections with lower scores to enhance the model's performance. To address overfitting on user and item representations, we implement a strategy involving polynomial distribution sampling to eliminate edges originating from recommended connections. By using the strategy, we effectively enhances the generalization and robustness. We use a predefined feature fusion algorithm to acquire the attempt-specific nodes of the item as contrastive embeddings. We adopt iteratively LightGCN to represent the $G_{u,i}$:

$$emb_u^l = \sum_{v \in \tau_u} \frac{x_v^{l-1}}{\sqrt{|\tau_v||\tau_u|}}; emb_v^l = \sum_{v \in \tau_u} \frac{x_u^{l-1}}{\sqrt{|\tau_u||\tau_v|}} \tag{15}$$

After obtaining the acquisition of the representation for the user-item interaction view, we aggregate the representations of each layer to obtain the final comparison embedding. Then, we feed enhanced comparison subgraphs into an MLP to further map them into the same latent space:

$$emb_u^* = \sigma(x_{v_i}^{*\top} \mathbf{W_1^*} + b_1^*)^\top \mathbf{W_2^*} + b_2^* \tag{16}$$

where $\mathbf{W}$ and $b$ are learnable weights and deviations. Furthermore, to ensure alignment of item representations across different views and enhance the robustness of the model, we incorporate a modified InfoNCE loss for optimization. We set $\kappa = \exp(s(emb_{v_i}^*, emb_{v_j}^*)/\beta)$, and $\gamma = \exp(s(emb_{z_i}^*, emb_{v_j}^*)/\beta))$. This loss will assign a random sample as a negative sample to each subgraph, and the specific formula is as follows:

$$\text{Loss}_{CL-SDKG} = \sum_{v \in V} -\log \frac{\kappa}{\sum_{z \in \{v, v', v''\}} (\kappa + \gamma)} \tag{17}$$

In the comparison loss, $v'$ and $v''$ serve as negative samples randomly drawn from sampled items. The similarity measurement $s(\cdot)$ is defined as the cosine similarity of the normalized vector. The temperature hyperparameter $\beta$ regulates the difficulty of comparison compared to the target. A higher $\beta$ might make the comparison more relaxed, with even slight differences being treated as similar, while a lower $\beta$ would make the comparison more restrictive.

Furthermore, We have enriched the perceptual representation of the existing knowledge graph, while combining self-derived knowledge graphs and existing knowledge graphs to enhance the recommendation system. Similarly, for the primary task, we utilize $y'_{uv} = emb_u^\top emb_v$ as the prediction result for recommendation, $emb_u$ is the enhanced user vector and $emb_v$ is the enhanced project vector. To refine the model parameters, we apply the widely embraced Bayesian Personalized Ranking (BPR) loss. We employ a holistic loss function to optimize all losses:

$$\text{Loss}_{rec} = \sum_{(u,i,z) \in D} -\log \sigma(y'_{uv} - y'_{uz}) \tag{18}$$

$$\text{Loss} = \text{Loss}_{rec} + \text{Loss}_{mask} + \lambda_1 \text{Loss}_{kg} + \lambda_2 \text{Loss}_{CL-SDKG} \tag{19}$$

where $\text{Loss}_{kg}$ represents the loss corresponding to the knowledge graph, $\text{Loss}_{mask}$ represents the mask loss. $\lambda_1$ and $\lambda_2$ are weights associated with the loss of the knowledge graph and self-derived subgraphs, respectively. These weights control whether the model is more inclined to learn the representation of its nodes or the representation of the knowledge graph nodes, and the sum of the two is equal to 1. In addition, because $\text{Loss}_{mask}$ enables the model to reconstruct effective connections between relational contexts, we add a mask loss [43] in CL-SDKG.

## 4.4 Complexity Analysis

In this section, we analyze the time complexity of the three main modules in our CL-SDKG. (1) For knowledge aggregation enhanced recommendation, we require the $O(|G_{kg}|d)$ to calculate rationale weighting and enhance the representation of KG, where $|G_{kg}|$ is the number of knowledge triplets in KG. (2) For variational graph reconstruction. CL-SDKG takes $O((|U| + |V|)Ld)$ to update GNNs, where $L$ is the number of GNN layers. This module also requires $O(2|V|dN + 2|U|)$ to generate self-derived subgraphs, where $L$ represents the number of non-zero elements in the adjacency matrix. Besides, enhancing the representation of the self-derived subgraphs incurs a cost of $O(|G_{kg}| + |V|d)$. (3) For contrastive learning of self-derived subgraphs in the user-item view, the module costs the $O((|emb_{u1}^*| + |emb_{u2}^*|)d)$.

## 5 EXPERIMENTS

## 5.1 Experimental Settings

*5.1.1 Datasets.* To thoroughly validate the effectiveness of CL-SDKG and ensure a comprehensive evaluation of diversity, we employed three distinct real-world datasets: Last FM, focusing on music recommendations, MIND for news recommendations, and Alibaba-iFashion, which caters to shopping recommendations. Furthermore, to enhance the performance of recommendation systems and mitigate noise introduced by knowledge graphs, we utilized the approach outlined in [43] to preprocess the dataset. Table 1 summarizes statistical details regarding user-item interactions and knowledge graphs for the three evaluation datasets.

**Table 1: Statistics of Three Evaluation Datasets.**

| Statistics | Alibaba-iFashion | Last-FM | MIND |
|---|---|---|---|
| Users | 114,737 | 23,566 | 100,000 |
| Items | 30,040 | 48,123 | 30,577 |
| Interactions | 1,781,093 | 3,034,796 | 2,975,319 |
| Density | 5.20E-04 | 2.70E-03 | 2.2E-4 |
| Knowledge Graph | | | |
| Entities | 59,156 | 58,266 | 24,733 |
| Relations | 51 | 9 | 512 |
| Triplets | 279,155 | 464,567 | 148,568 |

*5.1.2 Baseline Models.* In our experiments, we select 14 advanced recommendation algorithms as baselines for performance comparison. The description of these baselines is as follows.

**Collaborative Filtering Method Recommendation**.

- **BPR** [28]. It is an optimization standard directly aimed at personalized ranking, which obtains the estimator of the maximum posterior distribution through Bayesian analysis.
- **NCF** [11]. It is a universal framework rooted in neural network collaborative filtering. Multi-layer perceptrons are employed to learn the interaction function between users and items.
- **GC-MC** [3].It is a graph autoencoder framework based on bipartite interactive graph differentiable message passing. This algorithm utilizes message passing to generate potential features of users and item nodes.
- **LightGCN** [10]. It is a streamlined Graph Convolutional Network (GCN) model, obtaining user and item representations through linear propagation on the user-item interaction graph. The final representation is determined by the weighted sum of representations learned across all layers.
- **SGL** [29]. It is a multi-to-many seq2seq architecture trained using MNMT targets, which has excellent scalability for tasks such as cross-language AMR parsing.

**Embedding-based Knowledge-aware Recommendation**.

- **CKE** [51]. It is an integrated framework that utilizes collaborative learning to filter latent representations and semantic representations of items in a knowledge base.
- **KTUP** [4]. It considers various preferences when translating users into items, and then combines multiple transmission schemes with KG to complete model joint training.

**GNN-based KG for Recommenders**.

- **KGNN-LS** [33]. It entails a knowledge-aware Graph Neural Network incorporating label smoothing regularization. The knowledge graph transforms into a user-specific weighted graph, and a graph neural network is then applied to compute personalized item embeddings.
- **KGCN** [34]. It is a comprehensive framework designed to efficiently capture the correlation between items by extracting pertinent attributes from a knowledge graph.

- **KGAT** [36]. It pertains to knowledge graph attention networks, explicitly modeling high-order connections in the knowledge graph in an end-to-end manner.
- **KGIN** [37]. It represents a novel information aggregation scheme that recursively integrates relational sequences of distant connections. This algorithm extracts valuable information related to user intent and encodes it into representations of both users and items.

**Self-Supervised KG for Recommenders.**

- **MCCLK** [57]. It considers three graph views for knowledge graph recommendation: a global-level structural view, a local-level collaborative view, and a semantic view.
- **KGCL** [44]. It is a universal knowledge graph contrastive learning framework. This framework can reduce the information noise in knowledge graph-enhanced recommendation systems.
- **KGRec** [43]. It integrates generative and contrastive self-monitoring tasks, providing recommendations through rational blocking.

*5.1.3 Evaluation Metrics.* We adopt commonly used Recall@20 and NDCG@20 as evaluation indicators to measure the performance of our CL-SDKG and baselines. Furthermore, for a fair performance evaluation, we partition the datasets into three segments for each algorithm. Specifically, 70% is allocated for training, 10% for tuning hyperparameters, and 20% for testing.

*5.1.4 Parameter Settings.* Using PyTorch as our deep learning framework, we have implemented several baseline models by utilizing either official or third-party code. To further refine our proposed algorithm, we performed a hyperparameter search on the weight ratios associated with self-derived knowledge graphs and external knowledge graphs. Moreover, we search the weight of $Loss_{CL-SDKG}$ in the range of {0.1,. . . ,0.5,. . . ,0.9}, search the temperature of the CL in the range of {0.1,. . . ,0.5,. . . 0.9}, and we set masking sizes in the range of {128, 256}, maintaining proportions from from {0.2, 0.4, 0.6, 0.8}. Additionally, for all GNN-based methods, the number of GNN layers is fixed at 2.

## 5.2 Overall Performance Experiments

As shown in Table 2, we present a comparison between our CL-SDKG and baselines on the three real datasets. Based on the results, we draw the following observations.

Our proposed CL-SDKG algorithm outperforms others in both Recall and NDCG evaluation metrics. This superiority can be primarily attributed to two factors. Firstly, we employ a variational estimator to estimate the distribution of each node and introduce noise, significantly enhancing the model's robustness during the reconstruction process. Additionally, the contrastive learning of self-generated subgraphs diminishes the model's reliance on Knowledge Graphs (KG), thereby placing greater emphasis on discerning meaningful connections between user-item interactions and more effectively capturing user interest representations.

The approach of combining knowledge graphs does not consistently outperform the Collaborative Filtering Method. It is evident that, in both datasets, CKE and KTUP demonstrate superior recommendation performance compared to BPR, NCF, GC-MC, and

**Table 2: The overall performance evaluation results**

| Model | Last-FM | | Alibaba-iFashion | | MIND | |
|---|---|---|---|---|---|---|
| | Recall | NDCG | Recall | NDCG | Recall | NDCG |
| BPR | 0.0690 | 0.0582 | 0.0822 | 0.0501 | 0.0385 | 0.0253 |
| NCF | 0.0699 | 0.0615 | 0.0506 | 0.0276 | 0.0308 | 0.0237 |
| GC-MC | 0.0709 | 0.0631 | 0.0845 | 0.0502 | 0.0386 | 0.0261 |
| KGCL | 0.0905 | 0.0769 | 0.1146 | 0.0719 | 0.0399 | 0.0247 |
| LightGCN | 0.0738 | 0.0647 | 0.1058 | 0.0652 | 0.0419 | 0.0253 |
| SGL | 0.0879 | 0.0775 | 0.1141 | 0.0713 | 0.0429 | 0.0275 |
| KTUP | 0.0865 | 0.0671 | 0.0976 | 0.0634 | 0.0362 | 0.0302 |
| KGNN-LS | 0.0881 | 0.0690 | 0.0983 | 0.0633 | 0.0395 | 0.0302 |
| KGRec | 0.0928 | 0.0792 | 0.1179 | 0.0739 | 0.0381 | 0.0279 |
| KGAT | 0.0870 | 0.0743 | 0.0957 | 0.0577 | 0.0340 | 0.0287 |
| KGIN | 0.0900 | 0.0779 | 0.1144 | 0.0723 | 0.0357 | 0.0225 |
| MCCLK | 0.0671 | 0.0603 | 0.1089 | 0.0707 | 0.0327 | 0.0194 |
| CKE | 0.0845 | 0.0718 | 0.0835 | 0.0512 | 0.0387 | 0.0247 |
| KGCN | 0.0879 | 0.0694 | 0.0983 | 0.0633 | 0.0396 | 0.0302 |
| **CL-SDKG** | **0.0939** | **0.0813** | **0.1192** | **0.0748** | **0.0433** | **0.0331** |

LightGCN. However, the performance of SGL stands out even more, precisely highlighting the limitations of Knowledge Graphs (KG) in recommendation tasks. This phenomenon is particularly pronounced in datasets with complex knowledge graphs and sparse interactions.

The knowledge-aware recommendation models based on GNNs outperform the embedding-based models. It is evident that, in comparison to BPR and NCF, the knowledge-aware recommendation model based on GNN exhibits superior performance. This advantage arises from the linear propagation modeling utilized in the embedding model, whereas GNNs can capture more intricate and high-order information within the knowledge graph.

In the comparison between contrastive learning models (such as MCCLK and KGCL) and unsupervised models (such as KGIN), no model consistently excels on all datasets. The inconsistency observed could stem from the limitations of random graph enhancement or visually handcrafted cross-view pairing. These methods may fall short in fully uncovering genuinely valuable knowledge graph information for encoding user interests.

### 5.3 Weight Experiment of KG and Self-Generated Subgraphs

To evaluate the influence of weights $\lambda_2$ and $\lambda_1$ associated with KG and self-generated subgraphs on our CL-SDKG, we vary the corresponding weight values within the range of $(0.2, 0.8)$.

As shown in Figure 2, it is observed that the impact of $\lambda_2$ and $\lambda_1$ on CL-SDKG is not very significant. And when $\lambda_2 = 0.6$ and $\lambda_1 = 0.4$, we can see that our CL-SDKG obtains the best performance. The model performs better when the weights of KG and self-derived knowledge graphs are evenly matched in contrastive learning, indicating good robustness. Additionally, when $\lambda_2$ is varied within the range of $(0.2, 0.4)$ and $(0.6, 0.8)$, there is no substantial improvement in the overall performance of the model. This observation precisely underscores the limitations of KG in optimizing recommendation tasks, affirming the effectiveness of our work.

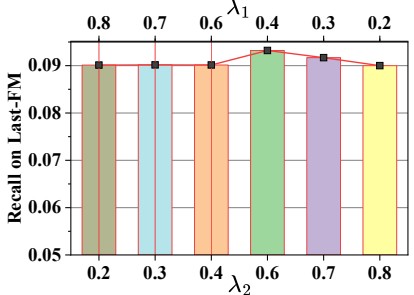

**Figure 2: The impact of $\lambda_1$ and $\lambda_2$ on CL-SDKG.**

### 5.4 Ablation Experiment

In this section, we conduct ablation experiments about losses to evaluate the impact of different losses on our CL-SDKG.

Based on the results shown in Table 3, $Loss_{rec}$ is the loss we only calculate for the recommendation task and use that loss to update the model. $Loss_{rec} + Loss_{mask}$ is the addition of the loss on the MASK task to update the model based on the previous ones. $Loss_{rec} + Loss_{mask} + Loss_{kg}$ is the experiment where we do not use $Loss_{CL-SDKG}$. And the last is the full version of our CL-SDKG. It is evident that when employing different losses to optimize the model, CL-SDKG with all losses in this paper achieves the best performance. In comparison to other variants such as our CL-SDKG without $Loss_{CL-SDKG}$, the full version of our CL-SDKG demonstrates the most effective recommendation performance on three real datasets. This underscores the significance of our proposed self-generating subgraph contrastive learning module, which significantly enhances the recommendation performance of our model.

### 5.5 Robustness Analysis

In this section, we delve into the robustness of our CL-SDKG from two perspectives: the number of layers and time parameters. Initially, we investigate the impact of graph inference layers on the performance. Subsequently, we further demonstrate the robustness of our CL-SDKG by examining the influence of temperature on contrastive learning. As shown in Figure 3, we compare our CL-SDKG with two state-of-the-art models based on the GNN.

**The impact of GNN-Layers**. As Figure 3 shows, in the Last-FM and Alibaba-iFashion datasets, we notice that as the number of graph inference layers increases, the performance of recommendations gradually improves and then reaches a plateau. Specifically, when the layer value is 1, the performance of all comparison methods is the worst. As the number of layers increases, other comparison methods vary widely, while our method consistently performs the best and demonstrates stable improvement. This suggests that when the number of layers is small, the graph network struggles to accurately estimate node distribution, while when the number of inference layers is excessively high it can lead to instability in estimation quality due to over-smoothing. It is evident that CL-SDKG outperforms other models overall and exhibits stronger robustness.

**The impact of Temperature**. Furthermore, as depicted in Figure 4, owing to the incorporation of contrastive learning in CL-SDKG, we investigated the influence of temperature on contrastive

Table 3: The impact of different losses on CL-SDKG

| CL-SDKG(Loss) | Last-FM | | Alibaba-iFashion | | MIND | |
|---|---|---|---|---|---|---|
| | Recall | NDCG | Recall | NDCG | Recall | NDCG |
| $Loss_{rec}$ | 0.0815 | 0.0705 | 0.1187 | 0.0745 | 0.0261 | 0.0145 |
| $Loss_{rec} + Loss_{mask}$ | 0.0842 | 0.0737 | 0.1183 | 0.0741 | 0.0371 | 0.0291 |
| $Loss_{rec} + Loss_{mask} + Loss_{kg}$ | 0.0921 | 0.0783 | 0.1178 | 0.0735 | 0.0374 | 0.0270 |
| $Loss_{rec} + Loss_{mask} + Loss_{kg} + Loss_{CL-SDKG}$ | **0.0939** | **0.0813** | **0.1192** | **0.0748** | **0.0433** | **0.0331** |

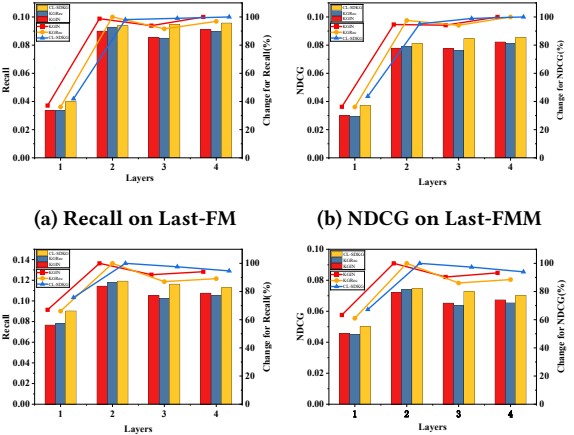

(a) Recall on Last-FM    (b) NDCG on Last-FMM

(c) Recall on Alibaba-iFashion (d) NDCG on Alibaba-iFashion

Figure 3: The impact of GNN-Layers

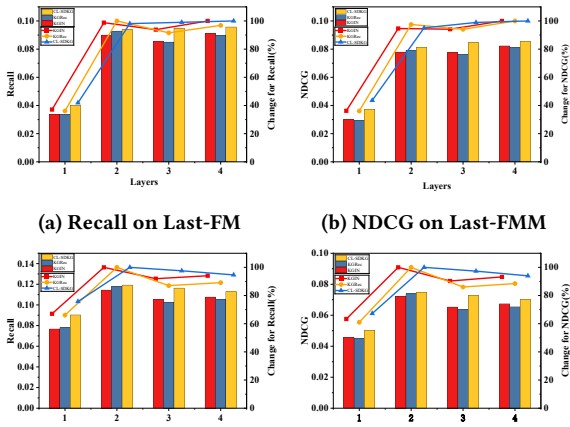

(a) Recall on Last-FM    (b) NDCG on Last-FMM

(c) Recall on Alibaba-iFashion (d) NDCG on Alibaba-iFashion

Figure 4: The impact of GNN-Layers

learning by comparing it with other models at varying temperature values. The results affirm the overall superiority of CL-SDKG compared to other models. Modifying the temperature values does exert a certain impact on the model's performance, but for CL-SDKG, the curve of change is relatively smooth, indicating it is not very sensitive to hyperparameter variations and possesses a considerable degree of robustness. In summary, CL-SDKG outperforms other advanced algorithms in terms of recommendation performance and robustness.

## 6 CONCLUSION

In this paper, we propose a novel self-derived knowledge graph contrastive learning (CL-SDKG) for recommendation. Our CL-SDKG employs a variational estimator to estimate the distribution of input graph nodes for deriving subgraphs that align with the original input graph. We utilize contrastive learning on user-item views to optimize the recommendation task. Experimental results on multiple datasets demonstrate the remarkable advantages of our CL-SDKG compared to state-of-the-art methods. In future work, we aim to further explore effective ways to integrate self-generated subgraphs with knowledge graphs, such as through graph adversarial generative learning and hypergraphs. This direction holds the potential to offer more possibilities in combining self-generated subgraphs and knowledge graphs.

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
