# OpenReview forum: "Self-derived Knowledge Graph Contrastive Learning for Recommendation"
_acmmm.org/ACMMM/2024/Conference — MM2024 Oral_

### Official Review · Reviewer_h4oe · 2024-05-09

**Rating:** 6
**Confidence:** 3

**Summary:**

This paper introduces a novel framework for enhancing recommendation systems using a self-derived Knowledge Graph (KG) and Contrastive Learning (CL) termed as CL-SDKG. The approach leverages variational graph reconstruction to estimate the distribution of user-item node pairs, generating multiple self-derived KGs that focus on reducing dependencies and enhancing user-item interaction analysis. Comprehensive experiments on three public datasets demonstrate that CL-SDKG outperforms standard baselines, showcasing its effectiveness in improving the robustness and accuracy of recommendations.

**Strengths:**

1. Innovation: The paper proposes a novel recommendation system framework, namely self derived knowledge graph contrastive learning (CL-SDKG), which combines variational graph reconstruction technology and contrastive learning to generate subgraphs that are consistent with the original input graph structure. The paper uses variational autoencoder (VAE) to estimate the Gaussian distribution of nodes and edges, achieving graph reconstruction. This method is theoretically reasonable and well founded. In addition, the robustness of the model was further improved through contrastive learning.
2. Clear structure: The methodology in the paper is based on the current research frontiers of Graph Neural Networks (GNN) and Knowledge Graph (KG), with clear technical details, clear and correct algorithm processes, and thorough consideration of details.
3. Full evaluation: The author conducted experiments on three different public datasets, covering a wide range of application scenarios, to verify the effectiveness of the proposed method and its practicality.
4. Research motivation: The article points out that existing knowledge aware recommendation models rely on the quality of knowledge graphs. Even worse, the model may overfill the KGs with information unrelated to recommendations.

**Limitations:**

1. Related work can be further introduced in the field of combining graph contrastive learning and knowledge graphs
2. The color matching in the experimental renderings can be improved
3. The fluency of English can be further enhanced

**Suitability:**

2

---

### Official Review · Reviewer_Jjny · 2024-05-21

**Rating:** 4
**Confidence:** 3

**Summary:**

This paper proposes a model (CL-SDKG) that uses variational graph reconstruction technology to improve the quality of knowledge graphs in GNN-based recommendation methods. In addition, it combines contrastive learning to enable CL-SDKG to model a more robust entity representation and enhance the ability to capture the interaction relationships between different items and entities.

**Strengths:**

1. Through variational graph reconstruction technology, the original graph structure is preserved as much as possible while enhancing the node representation ability in the graph.
2. Enhance GNN robustness through contrastive learning.

**Limitations:**

1. How is the time complexity of the proposed model compared to the baseline models?
2. How does the model's recommendation performance change if the self-derived knowledge graph proposed in this article is not used?
3. What is the `Rationale-aware Heterogeneous Knowledge Aggregation` mentioned in Figure 1?
4. In  robustness analysis section, only parameter experiment analysis is carried out. it does not verify the contribution mentioned in "it can effectively reduce the noise information brought by knowledge graphs and reduce excessive dependence on knowledge graphs”
5. Issues of Expression
   1) How to obtain $Loss_{mask}$?
   2) As defined in `PRELIMINARIES`, $\omega$ is the learnable parameter of the model, so what does $\omega$ in formula (14) mean?
6. Some Typos:
   1) Case issues: `We` in line 527, and `Our` in line 276;
   2) Punctuation issue: `.` in line 558 should be `,`;
   3) Statement consistency issue: Line 299 is expressed as `potional space`, while line 506 is called `latent space`;
7. Formula issues:
   1) Why is Q in superscript and k in subscript in $W^Q$ and $W_k$ in formula (1)? Is there any special difference between them?
   2) Is `v` in line 487 the same as `v` in formula (13)? (The font is different)

**Suitability:**

2

---

### Official Review · Reviewer_zix1 · 2024-05-24

**Rating:** 5
**Confidence:** 3

**Summary:**

The authors propose a Self-Derived Knowledge Graph Contrastive Learning framework (CL-SDKG) to enhance recommendation systems. The framework employs the variational graph reconstruction technique to generate self-derived Knowledge Graphs (KGs) by estimating the Gaussian distribution of user-item nodes. These self-derived KGs provide robust perceptual representations and allow models to focus on user-item interactions, minimizing the negative effects of dependencies from conventional KGs. Additionally, contrastive learning is applied to these self-derived KGs to enhance the robustness of the framework. Experiments conducted on three public datasets demonstrate that CL-SDKG outperforms state-of-the-art baselines.

**Strengths:**

The paper is easy to follow;
The tecnical contribution is clear.

**Limitations:**

I find the following aspects need to be addressed in the current version:
	Section 3 is too short; please consider merging it with Section 4.
	The paper contains formal symbolization errors, where vectors are not bold, such as u_i and v_i in Section 4 METHODOLOGY.
	Could not find the calculation formula for 〖Loss〗_mask. Please provide the corresponding formula.
	In Section 5.1.1, please add an explanation of the dataset preprocessing methods to facilitate reader understanding.
	In Section 5.2, please consider using underscores to label suboptimal baseline methods for easier comparison of performance across different methods.
	The field datasets used in this paper and the code is not available to the research community. Hence, the results are rather hard to reproduce.
   Consider adding more evaluation metrics.

**Suitability:**

2

---

### Official Review · Reviewer_Gr1b · 2024-05-24

**Rating:** 6
**Confidence:** 3

**Summary:**

In this paper, a self-derived Knowledge Graph Contrastive Learning Framework (CL-SDKG) is proposed to enhance the recommendation system. The variational graph reconstruction technique is used to estimate the Gaussian distribution of user item nodes. Finally, the contrastive learning is applied to KG to improve the robustness. And good results have been achieved in multiple dataset experiments.

**Strengths:**

1. In the process of generating knowledge graphs, the authors innovatively generate multiple knowledge graphs, which can reduce the over-reliance on knowledge graphs and reduce the impact of noise by estimating the consistency of the structure. It can solve the problem of over-reliance on graphs in the past.
2. The article is full of experiments and analysis of inter-complexity analysis, with rigorous logic and good writing.

**Limitations:**

1. Please add a more detailed explanation about Figure 1 for better understanding.
2. Why did you choose a Gaussian distribution and have you tried other distribution in experiments?
3. Is the space complexity too high when generating multiple graphs due to memory space resources?
4. The effectiveness of generating subgraphs can be expressed with visualization analysis.

**Suitability:**

2

---

### Meta-Review · Area_Chair_Njx3 · 2024-06-29

**Recommendation:** Accept (Oral)
**Confidence:** 5

**Metareview:**

All the reviewers agree that the paper is a high-quality paper. After reading the paper, I am confident that the paper is above the bar. Thus I recommend acceptance.